# STRUCTURE-AWARE DOMAIN KNOWLEDGE INJECTION FOR LARGE LANGUAGE MODELS

## ABSTRACT

This paper introduces a pioneering methodology, termed *StructTuning*, to efficiently transform foundation Large Language Models (LLMs) into domain specialists. It significantly reduces the training corpus requirement to a mere **0.3%**, while achieving an impressive **50%** of traditional knowledge injection performance. Our method is inspired by the educational processes of human students, particularly how structured domain knowledge from textbooks is assimilated and subsequently applied to tackle real-world challenges through specific exercises. Based on this, we propose a novel two-stage strategy for knowledge injection and alignment: *Structure-aware Continual Pre-Training* (SCPT) and *Structure-aware Supervised Fine-Tuning* (SSFT). In the SCPT phase, we automatically extract the domain knowledge taxonomy and reorganize the training corpora, enabling LLMs to effectively link textual segments to targeted knowledge points within the taxonomy. In the SSFT phase, we explicitly prompt models to elucidate the underlying knowledge structure in their outputs, leveraging the structured domain insight to address practical problems. Our ultimate method has undergone extensive evaluations across model architectures and scales, using closed-book question-answering tasks on LongBench and MMed-Bench datasets. Furthermore, we have investigated the scalability of structure-aware knowledge injection across varying sizes of training corpora, which lays a foundation for scaling up our StructTuning for stronger domain-specific LLMs with comprehensive data utilization. Code is available at this anonymous URL: https://anonymous.4open.science/r/StructTuning/.

## 1 INTRODUCTION

Large language models (LLMs) have recently seen extensive deployment across various applications (Vaswani et al., 2017; Achiam et al., 2023; Jiang et al., 2023; Bi et al., 2024). When adapting foundational models (*e.g.*, Llama series (Touvron et al., 2023a;b; Dubey et al., 2024)) for specialized AI assistants in distinct domains (Qiu et al., 2024; Guo et al., 2024), developers usually employ two techniques to enhance LLMs' proficiency: retrieval-augmented generation (RAG) (Lewis et al., 2020) and domain knowledge injection (Gururangan et al., 2020). While RAG effectively utilizes an external knowledge base to augment information, the retrieval process's inherent noise poses challenges to generating reliable responses, especially in scenarios requiring logical reasoning where there is a semantic gap between the user's query and the knowledge base.(Zhang et al., 2023; Chen et al., 2023). Thus, another avenue of research focuses on injecting new knowledge into models via training techniques (Gu et al., 2021; Hu et al., 2021; Mecklenburg et al., 2024).

Continual pre-training (Sun et al., 2020; Ibrahim et al., 2024) has been preferred for integrating new, domain-specific knowledge into existing LLMs (Cui et al., 2023; Wang et al., 2023b; Qiu et al., 2024). Nevertheless, it often entails resource-intensive auto-regressive training on billions of tokens from the internet to learn fragmented knowledge points, rather than absorbing structured knowledge from a few domain-specific textbooks (Jin et al., 2020). For example, MMedLM (Qiu et al., 2024) curates 25.5B tokens to derive a medical model, and DeepSeek-Coder (Guo et al., 2024) uses 2T tokens for coding adaptation. The common failure to learn effectively from limited textbook content has been attributed to insufficient data diversity (Zhu & Li, 2023a), which however violates the observation during the human education process in Fig. 1: students gain knowledge by sequentially studying from textbooks, reviewing knowledge points and structures, and applying this knowledge through proper

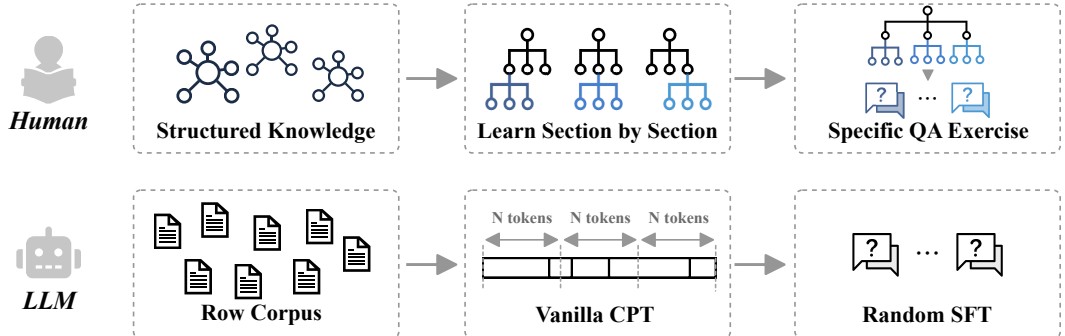

Figure 1: **Discrepancy between human learning process and vanilla LLM adaptation paradigm**. Human students learn structured knowledge through textbooks section by section, with particular exercises on related knowledge points. Traditional LLM adaptation involves continual pre-training on data chunks from randomly concatenated text segments, with aimless supervised fine-tuning for conversation alignment. The inherent property of structured knowledge is ignored.

exercises. In this process, all the new data to learn are textbooks (structured content) and exercising examples (question-answering pairs), and students just adopt their world knowledge to memorize, understand, and apply the knowledge to become domain experts (Krathwohl, 2002; Yu et al., 2023).

Inspired by this, we propose to inject the domain knowledge from textbooks into LLMs, as educating a human student, through a novel two-stage training strategy: *Structure-aware Continual Pre-Training* (SCPT) and *Structure-aware Supervised Fine-Tuning* (SSFT).

In the SCPT stage, we argue that high-quality textbook data can adequately infuse the domain knowledge (Gunasekar et al., 2023), where the organization of training corpora is crucial. In conventional paradigms, as illustrated in Fig. 1, text corpora are simply concatenated and divided into chunks of 2048 (Qiu et al., 2024) or 4096 (Guo et al., 2024), while the inherent structure of the texts (*e.g.*, catalogs of textbooks) is disregarded. Instead, we propose an automatic approach to maintain each chunk's knowledge structure. We view each chunk as a knowledge point, and employ advanced LLMs to efficiently extract domain knowledge taxonomy from the corpus, bypassing the need for manual annotation. Subsequently, LLMs are trained to predict the textual content (corresponding to the knowledge point) *under the condition of* the knowledge path within the domain structure, linking individual training chunks with the entire knowledge architecture. Finally, models are asked to memorize the knowledge structure to review the whole domain knowledge system.

In the SSFT stage, the goal shifts from knowledge injection to enabling LLMs to recall and utilize their acquired knowledge to tackle real-world challenges. We explicitly elicit knowledge paths in LLMs' responses, as a beacon for models to targeted information retrieval or logical reasoning for reliable responses. To this end, we derive a scalable strategy to generate question-answer pairs as practice exercises by powerful LLMs such as GPT4 (Achiam et al., 2023) or LLaMA3 (Dubey et al., 2024). In the scenarios with existing QA pairs like MMedBench (Qiu et al., 2024), we retrieve related knowledge structure and content, instructing LLaMA3 to provide explanations from questions to answers based on the knowledge paths. For datasets lacking specific QA samples like LongBench (Bai et al., 2023), we randomly select knowledge paths from the domain taxonomy and prompt LLaMA3 to craft question-answer-explanation triplets for training exercises.

Our ultimate approach, termed *StructTuning*, outperforms conventional methods in domain knowledge injection by emulating human learning processes through SCPT and SSFT phases. We extensively evaluate StructTuning's effectiveness across different model architectures and sizes. For domain-adapted language models, we first examine their capability to recall the injected knowledge through open-ended QA on the LongBench (Bai et al., 2023) dataset, then assess their application of this knowledge in addressing real-world issues via multiple-choice QA on MMedBench (Qiu et al., 2024). Both evaluations underscore the superiority of StructTuning. Remarkably, we achieve a **50%** improvement in knowledge injection compared to the SOTA MMedLM2 in the medical domain, using merely **0.3%** of the training data requirement. And StructTuning illustrates the potential of comparable performance with only 5% of training costs. These findings reveal the scalability of our method for enhancing domain-specific AI assistants with further comprehensive data utilization.

Our contribution is summarized as follows:

- We proposed a novel two-stage training strategy, SCPT and SSFT, to inject domain knowledge into LLMs by preserving and utilizing the inherent structure of the training corpus.
- We developed a scalable data construction framework to generate structure-aware training samples from original corpora, so as to facilitate the SCPT and SSFT stages.
- We conducted extensive investigations on our StructTuning strategy on various data and model settings, and comprehensively illustrate our superiority in knowledge injection.

## 2 RELATED WORK

**Domain Adaptation for Large Language Models**. While pre-trained LLMs possess promising capabilities, their performance is often hampered by the scope and recency of their training data, which particularly affects smaller models in downstream applications (Zhao et al., 2023; Wang et al., 2023a). Continual Pre-Training (CPT) addresses this by perpetually updating a pre-trained model with domain-specific content (Sun et al., 2020; Xu et al., 2023b). , with parameter-efficient tuning methods devised to curtail training costs (Hu et al., 2021; Liu et al., 2024c). To keep pace with the latest information, models can be fine-tuned with supervised instruction-response pairs (SFT), thus staying current with the advancing knowledge landscape (Mecklenburg et al., 2024; Qiu et al., 2024). Existing literature confirms that combining CPT and SFT is effective for LLMs to remain precise and up-to-date in dynamic fields like law (Cui et al., 2023; Nguyen, 2023) , finance (Wu et al., 2023; Li et al., 2024), medicine (Wang et al., 2023b; Qiu et al., 2024), and coding (Roziere et al., 2023; Guo et al., 2024). Our study builds upon this CPT-SFT framework, innovating with SCPT-SSFT strategies to efficiently and effectively infuse domain knowledge with the inherent structure hierarchy.

**Structure-aware Knowledge Aggregation**. Knowledge structure has been widely explored in the recent LLM community. A branch of researchers follows the conventional paradigm to extract entity-relation-entity triplets from texts to construct knowledge graphs (Pan et al., 2024), to enhance LLMs's factual knowledge and logical reasoning by feature aggregation (Liu et al., 2020; Zhang et al., 2022), prompt engineering (Wen et al., 2023; Wang et al., 2023c), information searching (Logan IV et al., 2019; Wu et al., 2022), training data synthesis (Tang et al., 2024), etc. In these cases, each node corresponds to either a specific entity or an abstract concept, lacking the capability to present an informative and self-contained *knowledge point*. Some works have recently related a piece of descriptive text to a knowledge point, and constructed the knowledge structure for LLMs' retrieval-augmented generation (Sarthi et al., 2024; Dong et al., 2024), where the top-to-down retrieval provides precise information-seeking paths along the knowledge structure. In this paper, we extend the structure-aware knowledge aggregation to LLMs' training phase, injecting the whole domain knowledge structure into LLMs' by linking the training samples into corresponding knowledge points and reasoning paths.

**Data Augmentation and Synthesis**. Due to the lack of high-quality datasets, data augmentation has emerged as a promising solution to mimic real-world patterns (Liu et al., 2024b). Traditional methods aim to artificially expand the training dataset size (Xu et al., 2023a; Mukherjee et al., 2023) or generate entirely new samples that could help models learn better or adapt to specific tasks (Tang et al., 2024). Yet, they often overlook the structured nature of domain knowledge, and the aimlessly generated samples may also lack diversity (Ovadia et al., 2023; Mecklenburg et al., 2024), leading to potentially suboptimal training outcomes when applied for domain adaptations (Mecklenburg et al., 2024; Tang et al., 2024). By contrast, our SSFT design is an innovative departure to address the challenge of retaining and utilizing the structured knowledge inherent in domain-specific content.

## 3 METHODOLOGY

Fig. 2 depicts our StructTuning methodology to inject domain knowledge into pre-trained LLMs using the inherent knowledge structure. With curated domain corpora (typically a few textbooks), we first develop an automatic approach to extract the knowledge structure, and associate text chunks to corresponding knowledge paths and points (Sec. 3.1). Then, we design a two-stage training strategy to inject the highly structured domain knowledge into language models by mimicking the human education process, comprising the SCPT (Sec. 3.2) and SSFT (Sec. 3.3) techniques.

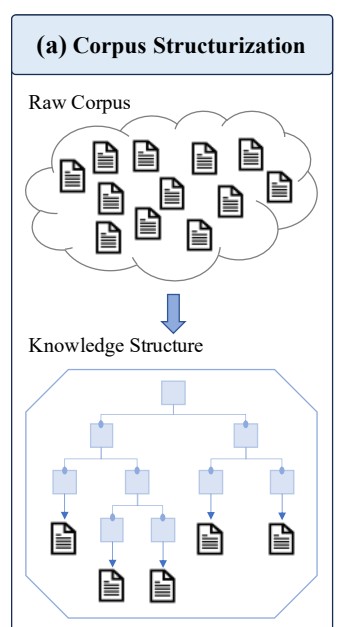
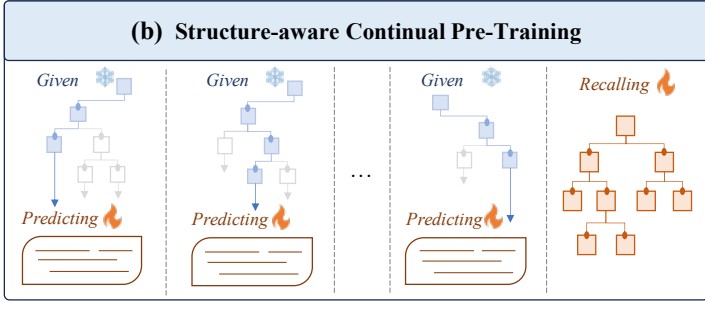
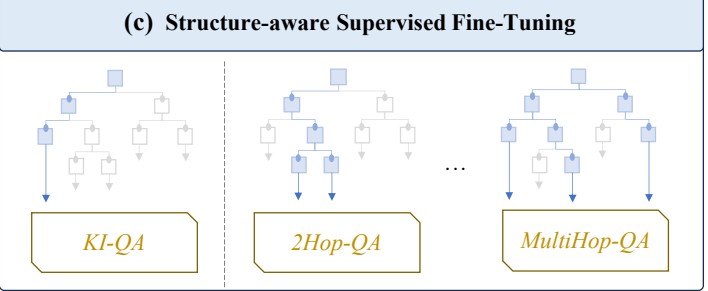

Figure 2: **Framework for structure-aware knowledge injection**. We extract the inherent knowledge structure in the training corpus, and associate training chunks to corresponding knowledge points. Models are continually pre-trained on data chunks in the condition of the knowledge structure, and fine-tuned with supervised QA samples to elicit their learned knowledge to solve real-world questions (including knowledge-intensive (KI) QA, 2- or multi-hop QA, *etc.*).

## 3.1 AUTOMATIC EXTRACTION OF KNOWLEDGE STRUCTURE

For web-crawled corpus, previous data pre-processing focuses on quality assessment for individual documents (Bi et al., 2024), while the meta-info of knowledge structures (*e.g.*, the table content for a textbook) is usually neglected or filtered out, and all we have are those sequentially arranged text segments (*e.g.*, page-by-page-chunked content). As shown in Fig. 2 (a), we aim to extract (or, recover) the knowledge structure from the raw corpus for subsequent domain knowledge injection.

First, we use spaCy[1] to split the content from a textbook at the paragraph-level, and merge the sentences to form training chunks within a maximum size (*e.g.*, 2048 tokens (Qiu et al., 2024)). After that, we prompt the advanced Llama3-70B (Dubey et al., 2024) model to summarize the title for each chunk, where the textual content with the abstractive title jointly contributes to a "knowledge point".

Then, we automatically aggregate knowledge points and extract the inherent structure hierarchy by leveraging advanced language models. Inspired by Liu et al. (2024a), we take the title list to instruct a specifically developed 7B model to identify the inherent knowledge structure within the text chunks. A prompt example is displayed in Fig. 3, and the detailed implementation is presented in Appendix B.1. In particular, Appendix B.1 and Appendix B.3 verify that our specialized 7B model can identify sufficiently precise knowledge structure for effective and efficient domain adaptation, as more powerful LLMs like LLaMA3-70B (Dubey et al., 2024) and GPT-3.5 (Brown et al., 2020) cannot bring significant enhancement while largely increase the inference costs.

Fig. 3 presents an example of the extracted structure, where we use the tree-like mindmap structure (Wen et al., 2023) to present the knowledge taxonomy from a textbook. And Fig. A1 showcases how to deal with non-textbook data. The whole process does not involve human annotation, which reduces the cost and makes our method scalable for larger domain training corpora.

After automatically extracting the domain knowledge structure and associating the original training chunks to related knowledge points, we delve into injecting domain knowledge through structure-aware continual pre-training (SCPT) and structure-aware supervised fine-tuning (SSFT).

---

[1]https://github.com/explosion/spaCy

**Knowledge Extraction Instruction**

Analyze the given content to extract and represent the intrinsic semantic hierarchy systematically. You should summarize the central theme and identify the core aspects of the discussion. For aspects with additional layers, delineate "SubAspects" and repeat as necessary for complex structures.

## Content
{*title_list*}

## Analysis

**Knowledge Structure**

**Biochemistry**
├── Overview of lipoprotein metabolism, hormone synthesis…
├── Lipoprotein Metabolism
│   ├── Lipid Metabolism and Cholesterol Transport
│   ├── Steroid Hormones: Synthesis, Regulation
│   └── Lipoprotein Metabolism and Hormone Synthesis
├── Steroid Hormones
│   ├── Nitrogen Metabolism: Amino Acid Catabolism
│   └── Pancreatic zymogen activation
…

Figure 3: **Left**: prompt template to extract hierarchical knowledge structures from the given content. **Right**: example of the extracted knowledge structure presented by a mindmap format.

## 3.2 STRUCTURE-AWARE CONTINUAL PRE-TRAINING

In conventional knowledge injection methods, training corpora are randomly concatenated and chunked into text segments without distinguishing the original content, leading to the fact that models can only absorb domain knowledge that is emergent in the data diversity (Ovadia et al., 2023; Mecklenburg et al., 2024; Qiu et al., 2024). In this section, we present another solution to inject knowledge from limited pieces of textbooks by leveraging the highly abstractive and exhaustive domain knowledge structures for continual pre-training.

We first transform the knowledge structure into natural languages using the same mindmap template in Fig. 3, and prepend it to each training chunk, forcing LLMs to memorize the textual content (knowledge points) in the condition of the associated knowledge path in the structure hierarchy. We collected 20 diversified templates from GPT-4 (Achiam et al., 2023) to bridge mindmap structures and training chunks, one of which is displayed in Fig. 4, and the full templates are presented in Fig. A5. The prepended mindmap, as well as the template, does not produce autoregressive loss. Losses are only calculated in the *content* part. Formally, we turn the original language modeling in vanilla CPT to conditioned modeling (Keskar et al., 2019) in our SCPT stage:

**SCPT chunk example**

In the realm of {*field*}, a conceptual mindmap is depicted using a tree-like structure to represent hierarchical relationships and thematic branches:

{*mindmap*}

Within this organized layout of {*field*}, the detailed subsection on {*section*} is described as:

{*content*}

Figure 4: Example of prompt templates to bridge mindmap structure and textual contents.

$$p(\boldsymbol{x}^k) = \prod_{i=1}^{n} p(x_i^k | x_{<i}^k) \quad \Longrightarrow \quad p(\boldsymbol{x}^k | \boldsymbol{s}^k) = \prod_{i=1}^{n} p(x_i^k | x_{<i}^k, \boldsymbol{s}^k) \tag{1}$$

where $p(\boldsymbol{x}^k)$ models the probability distribution for the $k$-th chunk $\boldsymbol{x}^k = (x_1^k, \cdots, x_n^k)$ via the chain rule of probability (Bengio et al., 2000) on each token $x_i^k$, and $\boldsymbol{s}^k$ denotes the associated knowledge mindmap. Appendix B.4 extensively investigates the effectiveness of our SCPT strategy.

As illustrated in Fig. 2 (b), after traversing the $m$ knowledge points in extracted structures, models are asked to recall the whole knowledge hierarchy, *i.e.*, to model the composed probability distribution:

$$p(\bar{\boldsymbol{s}}) = \prod_{k=1}^{m} p(\boldsymbol{s}^k) \tag{2}$$

In SCPT, we mimic the human education process to inject knowledge into LLMs in a section-by-section manner, and replay the entire knowledge structure for the models to review and summarize the learned domain knowledge. These two steps iteratively alternate throughout training epochs.

Next, we will introduce how to teach LLMs to explicitly utilize their domain knowledge, which is learned in the SCPT stage, to solve practical problems by doing exercises with our SSFT technique.

**(a) Knowledge-Intensive QA Generation**    **(b) Multi-hop QA Generation**

Figure 5: **QA samples synthesized for SSFT**. We instruct Llama3-70B to generate (a) knowledge-intensive and (b) multi-hop questions and derive the diagnosis answers with explicit reasoning.

## 3.3 Structure-aware Supervised Fine-Tuning

In traditional knowledge injection paradigms, supervised fine-tuning aims to align the (continually) pre-trained models to interactive ChatBots through massive question-answering exercises (Cui et al., 2023; Qiu et al., 2024). However, most QA data augmentation strategies focus on enlarging the quantity and enhancing the diversity of training samples (Xu et al., 2023a; Mukherjee et al., 2023; Liu et al., 2024b), which neglects the nature of the highly structured domain knowledge. Therefore, our structure-aware supervised fine-tuning (SSFT) technique focuses on eliciting models' structured knowledge learned during the SCPT stage, adapting LLMs to interactive and reliable domain experts.

Fig. 2 (c) illustrates the idea of synthesizing SSFT samples guided by domain knowledge structures. Specifically, we use the random walk algorithm to create knowledge paths with 1 to $l$ branches in the original mindmap (the illustration of knowledge paths and branches is displayed in Fig. A2). For paths linking to a single knowledge point, we use the corresponding text content to prompt Llama3-70B (Dubey et al., 2024) to generate knowledge-intensive question-answering pairs. For paths with two or more branches, we prompt Llama3-70B with the knowledge path and textual contents to synthesize 2- or multi-hop QA samples, which require specific reasoning along the knowledge structure to derive from questions to answers. Fig. 5 presents several examples.

For every synthesized QA sample ($z$), we will prepend the relevant mindmap hierarchy to the answer, and add a CoT prompt in the question to construct another type of QA data ($z'$) for SFT alignment. This design explicitly elicits the learned knowledge in models' responses, teaching them how to apply the structured knowledge to address real-world problems. We use the two types of QA samples for training, as recommended by Qiu et al. (2024). During testing, we can either use the vanilla question as input to efficiently gather models' answers, or take the CoT prompt to probe to what extent LLMs can memorize and leverage the injected knowledge to answer the questions.

Integrating with SCPT and SSFT, our StructTuning approach translates into remarkable efficacy and efficiency in domain knowledge injection, as comprehensively evaluated in the following sections.

## 4 Experiments

We design a comprehensive evaluation of our StructTuning through several experiments on two benchmarks. First, we investigate the free-form question-answering task on the LongBench (Bai et al., 2023) dataset, in order to verify the *memorization and understanding* of injected knowledge (the answer can be directly found in training corpora). Then, we delve into the multi-choice question-answering task on MMedBench (Qiu et al., 2024), to explore how LLMs *apply* the injected knowledge in basic medicine to determine the real-world diagnosis for patients with logical reasoning.

### 4.1 Preliminary Investigation on Free-form Question-Answering

**Datasets and Tasks.** LongBench (Bai et al., 2023) is a multi-task benchmark tailored for open-book reading comprehension evaluation, where LLMs generate answers to given questions based on one or

Table 1: Recall evaluation of Open-Book QA (OBQA) and Closed-Book QA (CBQA) tasks on the LongBench (Bai et al., 2023) dataset. The best results are marked in **bold**, and the secondary results are marked with underlines. The backbone model is Llama2-7B (Touvron et al., 2023b).

| Task | Adaptation | SingleDoc-QA | | | MultiDoc-QA | | | | Average |
|------|-----------|--------|------|--------|------|-------|-------|------|---------|
| | | Qasper | MFQA | MFQAzh | HpQA | 2Wiki | Musiq | Duzh | |
| OBQA | - | **39.7** | 44.3 | 17.5 | 30.1 | 35.5 | 11.9 | 9.6 | 26.9 |
| CBQA | CPT+SFT | 20.7 | 35.3 | 20.6 | 29.9 | 32.1 | 18.9 | 12.0 | 24.2 |
| | SCPT+SFT | 18.8 | 42.5 | 17.7 | 35.7 | 36.4 | 20.5 | 15.3 | 26.7 |
| | SCPT+SSFT | 30.5 | **44.6** | **24.3** | **40.8** | **42.0** | **21.8** | **16.8** | **31.5** |

several input passages. To focus on knowledge injection, we turn the open-book evaluation into a closed-book QA task, where LLMs are trained on contextual passages and queried with questions only. We choose 7 subsets with 1,350 test examples from LongBench for single- and multi-document question-answering evaluation, and the remaining synthetic or code-orientated tasks are eliminated. More details are described in Appendix A.1.

**Evaluation Metrics.** To quantify the knowledge memorization degree, we mainly report the *recall* (Zhu & Li, 2023b) for models' outputs against ground-truth answers for the free-form QA tasks. In Appendix B.8 we also evaluate our method with the F1-score measure for a thorough comparison.

**Investigated Models.** For LongBench, we mainly investigate the knowledge injection to Llama2-7B (Touvron et al., 2023b) to compare the open- and closed-book QA performance.

**Implementation Details.** For closed-book QA, the Llama2-7B model is continually pre-trained on 10,476 passages for 3 epochs using a batch size of 128. The initial learning rate is 2e-5, decayed to 0 at the end of training with a cosine scheduler. We query Llama3-70B to generate 2,700 SSFT QA examples and remove those with over 0.5 F1-Score similarity to test samples to prevent data leakage (details are presented in Appendix A.1). Models are trained for 1 SFT epoch to avoid overfitting.

**Main Results.** First, we use a CoT instruction of "*Answer the question and explain why*" to set a baseline for Open-Book QA (OBQA). As shown in Tab. 1, it presents a moderate knowledge comprehending capability in SingleDoc QA tasks (*e.g.*, over 40% recall on Qasper and MFQA), and relatively poor knowledge coordination in MultiDoc QA subsets (*e.g.*, around 10% recall on Musiq and Duzh) due to the attention drift on longer inputs.

Then, we try to inject passage content into LLMs with a conventional CPT+SFT paradigm to benchmark the Closed-Book QA (CBQA) baseline. Note that we use two types of question templates (refer to Sec. 3.3) during training, and use the CoT version during testing to elicit models' memorized knowledge in their responses. However, Tab. 1 indicates such a knowledge injection approach is ineffective, as the knowledge recall is even lower than the OBQA baseline (*i.e.*, 24.2% *v.s.* 26.9%).

*Question*

Regarding Multilingual dialect detection and classification using deep learning, what models were used for dialect identification?

*Model Response*

Here is the knowledge structure regarding Multilingual dialect detection and classification using deep learning:
Multilingual dialect detection and classification using deep learning
├── Exploring multilingual dialect identification using deep learning techniques
├── Design and implementation of a neural network for language identification
├── Proposed method for multilingual dialect identification using CNNs and LSTMs
├── Evaluation and analysis of a dialect identification model using X-NMT architecture
└── Analysis of a multilingual dialect identification system using NMT models

The question asks about the specific models employed in the study for dialect identification. By recalling the passage's description of the "proposed method," we can identify that **BERT, RoBERTa, and XLM-RoBERTa** were used as these three architectures were combined to form the X-NMT model

*Ground-Truth*

BERT, RoBERTa, ELECTRA, and XLM-RoBERTa.

Figure 6: An example for structure-aware responses.

On the contrary, our SCPT strategy has already achieved a considerable knowledge recall of 26.7%, which is higher than the CPT-SFT paradigm of 24.2% and approaching the OBQA baseline of 26.9%. It implies our structure-aware continually pre-trained model has successfully associated the relevant passages with their entire knowledge structure for the given question, which provides the knowledge path to seek targeted information to derive the answer.

Furthermore, our SSFT technique explicitly teaches the model to recall the learned knowledge and answer the questions, which further improves the knowledge recall to 31.5% and even surpasses the Open-Book QA setting. The results indicate the vanilla SFT strategy can only regularize LLMs' response styles, while our SSFT could teach LLMs to utilize their knowledge (injected in the SCPT stage) to answer corresponding questions, as exemplified in Fig. 6.

Particularly, our method receives significant enhancements on MultiDoc QA subsets, which implies even though the total text content may exceed the models' attention window and influence the OpenBook-QA, we can still chunk the content into several pieces and successfully inject the knowledge points into LLMs meanwhile preserving the whole knowledge structure.

In addition, we also use lexical ROUGE-L (Lin, 2004) and semantic BERTScore (Zhang et al., 2020) to quantify the memorization of injected knowledge structures, by comparing the mindmap in models' responses (as Fig. 6 displays) with ground-truth answers. The results in Tab. 2 indicate a relatively good memorization of the injected knowledge mindmap, emphasizing the efficacy of our SCPT strategy.

Table 2: MindMap Recall

| F1-Score | BERTScore |
|---|---|
| 0.61 | 0.87 |

### 4.2 IN-DEPTH EVALUATION FOR MULTI-CHOICE QA APPLICATION

**Datasets and Tasks.** MMedBench (Qiu et al., 2024) is a multilingual medical multi-choice QA benchmark, with 45,048 QA pairs for adapting LLMs to medical experts and 8,518 for testing. We collect 76M textbook corpora from MedTextBooks (Jin et al., 2020) and MMedC (Qiu et al., 2024) for medical knowledge injection and use the training/test split from MMedBench for SFT/evaluation. We also curate another two sizes of training sets (with 30M and 132M tokens) to validate our method's scalability. Detailed setup is in Appendix A.2.

**Evaluation Metrics.** For multi-choice QA in MMedBench (Qiu et al., 2024), we follow the default setting to calculate the accuracy on six language subsets, as well as the averaged scores. Metrics are computed by lexical exact-matching on models' responses, rather than maximum token probabilities.

**Investigated Models.** For MMedBench, we extend the investigated LLMs across model scales and architectures including Llama2-7B/13B (Touvron et al., 2023b), InternLM2-7B (Zheng et al., 2024), and the recent Llama3-8B (Dubey et al., 2024). We also compare our knowledge-injected models with other popular domain-specified LLMs, such as MedAlpaca (Han et al., 2023), ChatDoctor (Yunxiang et al., 2023), PMC-LLaMA (Wu et al., 2024), and MMedLM (Qiu et al., 2024) models.

**Implementation Details.** We first train LLMs for 3 epochs on medical textbooks with a batch size of 128. Then, we ask Llama3-70B to create structure-aware explanations for existing 45K QA samples in MMedBench's training split and 33K extra entries by traversing extracted knowledge structures. Syntheses with overlapped options in the test set are removed. In the SFT phase, the learning rate is set as 1e-6 to avoid overfitting on such an amount of SFT samples, as suggested by Qiu et al. (2024).

**Main Results.** In Tab. 3, we present the overall performance across a series of LLMs on the testing split of MMedBench. The results demonstrate the promising enhancement achieved by our StructTuning technique, which translates into consistent improvements on the advanced InternLM2-7B (Zheng et al., 2024) and Llama3-8B (Dubey et al., 2024) models, and largely outperforms the previous domain-specific LLMs like PMC-LLaMA (Wu et al., 2024) and MedAlpaca (Han et al., 2023). Notably, our structure-aware knowledge injection approach, using merely 76M tokens curated from medical textbooks, achieves over **50%** performance (4.46% *v.s.* 8.71%, 2.57% *v.s.* 4.96%) against the state-of-the-art MMedLM (Qiu et al., 2024) method, which is trained on a huge MMedC (Qiu et al., 2024) corpora of 25.5B tokens. Our approach shows much more efficiency in transforming pre-trained LLMs into domain experts by structured-aware knowledge injection, *e.g.*, we only use 0.3% corpus to achieve comparable or even slightly better performance on English and Spanish subsets on top of Llama3-8B.

However, our method is currently unable to achieve 100% of knowledge injection efficacy against MMedLM, which may comprise two major reasons: (1) some knowledge in the training corpus has already been learned by the foundation model (*e.g.*, a slight decrease in the Chinese subset), and (2) knowledge in the curated 76M training data cannot cover all testing scenarios (*e.g.*, a considerable gap on the Japanese subset due to the absence of Japanese corpus, see Tab. A3). Therefore, we make a step to evaluate our approach's scalability on various corpora sizes in Fig. 7.

Table 3: Multiple-choice accuracy evaluation on MMedBench (Qiu et al., 2024). We report each model's accuracy across six languages separately, with "Average" denoting the mean score over six languages. We also compare the data requirement ("#Token") for medical knowledge injection.

| Model | English | Chinese | Japanese | French | Russian | Spanish | **Average** | #Token |
|---|---|---|---|---|---|---|---|---|
| ChatDoctor | 43.52 | 43.26 | 25.63 | 18.81 | 62.50 | 43.44 | 39.53 | - |
| PMC-LLaMA | 47.53 | 42.44 | 24.12 | 20.74 | 62.11 | 43.29 | 40.04 | - |
| MedAlpaca | 46.74 | 44.80 | 29.64 | 21.06 | 59.38 | 45.00 | 41.11 | - |
| Llama2-7B | 43.36 | 50.29 | 25.13 | 20.90 | 66.80 | 47.10 | 42.26 | - |
| InternLM-7B | 44.07 | 64.62 | 37.19 | 24.92 | 58.20 | 44.97 | 45.67 | - |
| InternLM2-7B | 57.27 | 77.55 | 47.74 | 41.00 | 68.36 | 59.59 | 58.59 +0.00 | - |
| InternLM2+MMed | **61.74** | **80.01** | **61.81** | **52.09** | **80.47** | **67.65** | **67.30** +8.71 | 25.5B |
| InternLM2+Ours | 60.80 | 79.19 | 50.75 | 45.34 | 75.39 | 66.85 | 63.05 +4.46 | **76M** |
| Llama3-8B | 63.86 | 78.23 | 48.24 | 50.80 | 71.48 | 64.15 | 62.79 +0.00 | - |
| Llama3+MMed | 66.06 | **79.25** | **61.81** | **55.63** | **75.39** | 68.38 | **67.75** +4.96 | 25.5B |
| Llama3+Ours | **66.77** | 77.44 | 53.27 | 51.61 | 74.61 | **68.49** | 65.36 +2.57 | **76M** |

**Approach's Scalability.** In Tab. 3, we use roughly 0.3% of the 25.5B tokens in MMedC Qiu et al. (2024) to evaluate knowledge injection methods. Here, we curate another two training corpora sizes: 30M and 132M, which take around 0.1% and 0.5% of 25.5B tokens respectively. We compare the vanilla CPT-SFT paradigm and our SCPT-SSFT strategy across those data settings The backbone LLM is InternLM2-7B (Zheng et al., 2024), and the training settings follow the main experiments. According to Fig. 7, our method consistently surpasses the vanilla paradigm by a large margin, emphasizing the efficacy and efficiency of domain knowledge injection. In particular, we can fit two performance-ratio scaling curves from the data points in Fig. 7 as:

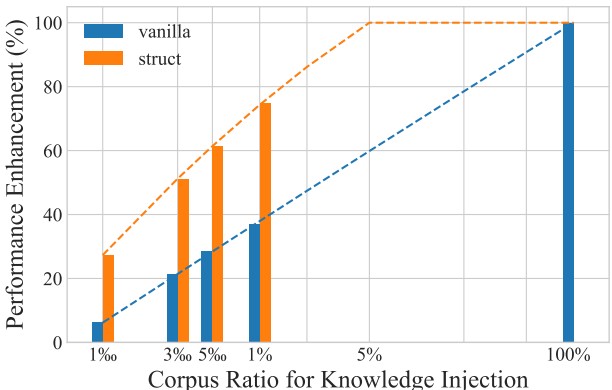

Figure 7: Comparison of knowledge injection approaches.

$$p_v \approx -0.04(\log r)^2 + 13.3 \log r + 100.0; \quad p_s \approx -1.11(\log r)^2 + 7.63 \log r + 133.0 \quad (3)$$

where $p_v$ and $p_s$ denote the relative performance enhancement (%) for vanilla and structure-aware knowledge injection, and $r$ refers to the corpus ratio. In Appendix B.2, we successfully use the fitted scaling law to predict the performance on 1% training corpus (around 250M CPT/SCPT tokens), which further enhances the reliability of this hypothesis. The scaling law indicates we can achieve comparable performance (100%) with only **5%** of the total training corpus, significantly reducing the costs for LLMs' domain adaptation. On the other hand, it also indicates our method may lead to 133% enhancement with a further 100% comprehensive data utilization. However, we have not empirically verified those predictions due to time and resource constraints, which we view as a temporal limitation of our current work and leave it to future investigations.

**Approach's Generalization.** In addition to the InternLM2 and Llama3 models in previous experiments, we also investigate the generalization ability of our SCPT+SSFT paradigm on Llama2 (Touvron et al., 2023b) model series. As shown in Tab. 4, our method leads to consistently significant improvements on 7B (+8.78%) and 13B (+6.17%) backbone models. The results further demonstrate the generalizability and scalability of our StructTuning strategy across model architectures and sizes.

**Ablation Studies.** To further validate the efficacy of StructTuning, we conduct a comprehensive ablation study with the English split of the MMedBench dataset. Specifically, we take Llama2-7B as the backbone model, select the English textbooks (Jin et al., 2020) (with 26M tokens) for vanilla and

Table 4: Structure-aware knowledge injection to Llama2 (Touvron et al., 2023b) model series.

| Model | English | Chinese | Japanese | French | Russian | Spanish | **Average** |
|---|---|---|---|---|---|---|---|
| Llama2-7B | 43.36 | 50.29 | 25.13 | 20.90 | 66.80 | 47.10 | 42.26 |
| **+Ours** | **49.41** | **65.15** | **36.68** | **35.21** | **69.14** | **50.62** | **51.04** |
| Llama2-13B | 51.37 | 57.97 | 32.66 | 25.08 | 69.92 | 52.99 | 48.33 |
| **+Ours** | **53.02** | **68.30** | **37.78** | **41.71** | **70.70** | **55.51** | **54.50** |

our structure-aware continual pre-training, and use different QA samples for supervised fine-tuning. In Tab. 5, "SFT" refers to vanilla SFT with 10K English QA samples provided in MMedBench's training split, "SSFT" indicates structure-aware SFT on 10K training data, where the questions are the same as "SFT" while the answers are enhanced with knowledge structure explanation by Llama3-70B, as described in Sec. 3.3. "SSFT*" includes another 8K structure-aware QA syntheses by Llama3-70B, consisting of 18K entries for training. The training hyper-parameters follow the main experiment.

Table 5: Ablation studies with Llama2-7B on the English subset of MMedBench.

| Adaptation | | English | Chinese | Japanese | French | Russian | Spanish | **Average** |
|---|---|---|---|---|---|---|---|---|
| - | SFT | 44.54 | 32.81 | **26.63** | 15.27 | 53.91 | 42.30 | 35.91 |
| CPT | SFT | 46.27 | 32.57 | 26.13 | 17.36 | 50.00 | 40.63 | 35.49 |
| **SCPT** | SFT | 46.50 | 32.14 | 20.10 | 18.17 | 53.91 | 39.97 | 35.13 |
| **SCPT** | **SSFT** | **49.96** | 32.63 | 22.11 | 17.52 | 51.17 | 41.28 | 35.78 |
| **SCPT** | **SSFT*** | 49.10 | **33.92** | 18.33 | **27.14** | **57.42** | **43.73** | **38.27** |
| RAG | | 38.12 | 29.22 | 22.61 | 23.34 | 53.91 | 36.47 | 33.95 |

In Tab. 5, the CPT+SFT paradigm does bring 1.73% improvement in the English split, while our SCPT technique with vanilla SFT presents a slightly higher accuracy of 46.50%. When combining SSFT with SCPT, our method immediately leads to a significant boost in the English split (49.96% *v.s.* 44.54%), demonstrating the efficacy and necessity of eliciting the learned structured knowledge to solve practical problems. Moreover, the supplemented 8K extra QA syntheses ("SSFT*") surprisingly enhanced the model performance on the other five subsets, which demonstrates the knowledge transferability across different languages (Lai et al., 2023; Qin et al., 2024). After training with SSFT, LLMs can actively utilize the knowledge injected in one language to solve the problem in another language, evidencing our superiority against the traditional SFT technique. In Appendix B.5, we follow Liu et al. (2024b) to randomly generate another 8K SFT pairs for further comparison, where our structure-aware SFT syntheses are verified to better enhance LLMs' knowledge application.

In addition, we observe that the commonly used RAG (Lewis et al., 2020) strategy does not bring significant advantages to the MMedBench evaluation. The main reason lies in the gap between the pre-training corpus (comprising official knowledge statements from textbooks) and evaluated QA samples (originating from practical diagnosis records). Knowledge injection by (S)CPT and (S)SFT shows more advantages in this situation. In-depth investigations can be found in Appendix B.7.

## 5 CONCLUSION

This work pioneers in incorporating structure-aware methodologies to enhance domain knowledge injection into large language models. Through a novel SCPT-SSFT paradigm, we have set a new precedent for adapting LLMs to specialized domains, and the promising and scalable results underscore the viability and potential of our method. We hope to inspire further research in efficient and effective domain adaptation, moving a step closer to models that can truly emulate human intelligence.

**Limitation.** A noteworthy limitation of our work is the under-explored full potential of our StructTuning strategy for knowledge injection. Although we have estimated a scaling law in Eq. (3), the data requirement to achieve 100% of the desired effectiveness remains unverified. Besides, our method introduces negligible data preprocessing costs to extract the domain knowledge structure. However, Appendix B.3 indicates our method can still reduce the overall adaptation cost to 10% since only a small part of training corpus is required. We will delve into the investigations in our future work.

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

## A   IMPLEMENTATION DETAILS

### A.1   DETAILED SETUP ON LONGBENCH

**Dataset Composition.** To focus on the investigation of knowledge injection, we choose 7 subsets from LongBench (Bai et al., 2023) across single- and multi-document QA tasks in English and Chinese, and the remaining synthetic or code-orientated tasks are eliminated:

- **Single-Doc QA.** For single-document QA, we take three subsets from LongBench: (1) *Qasper* (Dasigi et al., 2021), featured by question-answering over NLP technical papers and annotated by NLP practitioners; (2) *MultiFieldQA* (Bai et al., 2023), manually curated from multiple data sources and annotated by Ph.D. students; and (3) *MultiFieldQA-zh*, the Chinese split also provided by Bai et al. (2023), covering multiple Chinese scenarios. *MultiFieldQA* contains 150 Context-Question-Answer triplets to test, and the others adopted subsets include 200 pieces of test samples respectively.

- **Multi-Doc QA.** Multi-document QA requires LLMs to extract and combine information from multiple documents to derive the answer, which is generally more challenging than single-doc QA. We take four multi-hop QA datasets: (1) *HotpotQA* (Yang et al., 2018), containing 2-hop questions written by native speakers given two related paragraphs; (2) *2WikiMultihopQA* (Ho et al., 2020), involving up to 5-hop questions synthesized through manually designed templates on Wikipedia passages; (3) *MuSiQue* (Trivedi et al., 2022), carefully composed with up to 4-hop reasoning on an increased number of supporting and distracting context evidence; and (4) *Dureader* (He et al., 2017), developed based on Baidu Search and Baidu Zhidao and filtered by Bai et al. (2023) to reduce the data noise. Each subset has 200 test samples.

For data entries in Single-Doc QA, we extract the knowledge structure for each single passage; in Multi-Doc QA, we identify the knowledge structure across multiple passages for each test sample. There are ultimate 1350 *question-answer-passage(s)-(knowledge)structure* quadruples to evaluate knowledge injection approaches on LongBench.

**SFT DataSynthesis.** We compute the F1-Score between the answers of generated SFT data and original test data to avoid knowledge leakage at the SFT stage. During inference, when the model can generate correct answers (corresponding to specific knowledge points) that haven't been seen during the SFT stage, we can ensure the knowledge is injected at the CPT stage and SFT only enhances the instruction-following capability. In practice, merely 13 out of 2700 (around 0.5%) synthetic data have over 0.5 F1-Score and are thus filtered out from the SFT data.

In Tab. A1, we also statistic the semantic similarity (measured by BERTScore (Zhang et al., 2020)) between generated and evaluated questions and answers, and the results emphasize there is no knowledge leakage in the generated SFT data (they share poor semantic similarity across questions, answers, as well as QAs).

Table A1: Similarity statistics on synthetic SFT data and LongBench's test samples.

| Target | Question | Answer | Question-Answer |
|--------|----------|--------|-----------------|
| BERTScore | 0.277 | 0.106 | 0.093 |

### A.2   DETAILED SETUP ON MMEDBENCH

**Data for Evaluation.** The Multilingual Medical Benchmark (MMedBench) (Qiu et al., 2024) represents a comprehensive and diverse multilingual medical Question and Answering (QA) benchmark designed to evaluate models' capabilities of understanding and processing medical content.

MMedBench's robust dataset extends across 6 languages (*i.e.*, English, Chinese, Japanese, French, Russian, and Spanish) and 21 medical fields, which include, but are not limited to, Internal Medicine, Biochemistry, Pharmacology, Psychiatry, and many others. It provides 45,048 training pairs and 8,518 testing pairs for diverse learning and testing scenarios. The training split is specifically designed for domain-specific finetuning of large language models (LLMs), while the entire testing set allows for

a precise assessment of multi-choice question-answering performance. Statistics on six languages are displayed in Tab. A2. Notably, the benchmark includes scenarios where questions may have multiple correct answers (*i.e.*, in Japanese and French subsets), introducing additional complexity for the model evaluation process.

Table A2: Sample statistics on MMedBench.

| Split | English | Chinese | Japanese | French | Russian | Spanish | Total |
|-------|---------|---------|----------|--------|---------|---------|-------|
| Train | 10,178 | 27,400 | 1,590 | 2,171 | 1,052 | 2,657 | 45,048 |
| Test | 1,273 | 3,426 | 199 | 622 | 256 | 2,742 | 8,518 |

**Data for Continual Pre-Training.** To investigate high-quality domain knowledge injection for LLMs, we collect 18 English textbooks and 33 Chinese textbooks from the National Medical Board Examination in the USA and Mainland China, respectively (Jin et al., 2020). All collected textbooks are originally in PDF format and Jin et al. (2020) converted them into digital text via OCR and performed some clean-up pre-processing strategies to reduce the data noise. The English and Chinese textbooks count for around 26.1M and 21.5M tokens by Llama2 tokenizer (Touvron et al., 2023b). Then, we randomly sample an extra 28M textbook corpora from MMedC (Qiu et al., 2024) for the other languages (except the unavailable Japanese textbooks), resulting in a 76M token corpus for CPT. The statistics are displayed in Tab. A3.

Table A3: Sample statistics on training data. "†" and "‡" refer to different sizes.

| Stage | Ratio | English | Chinese | Japanese | French | Russian | Spanish | Total |
|-------|-------|---------|---------|----------|--------|---------|---------|-------|
| CPT | 0.3% | 6.9M | 8.8M | - | 4.1M | 4.9M | 5.4M | 30.1M |
| CPT | 0.1% | 26.1M | 21.5M | - | 8.1M | 10.3M | 10.1M | 76.1M |
| CPT | 0.5% | 35.9M | 27.2M | 4.5M | 14.0M | 24.9M | 26.1M | 132.6M |
| CPT | 1.0% | 44.1M | 39.8M | 34.2M | 45.1M | 47.9M | 38.4M | 249.5M |
| SFT | - | 18.8K | 39.1K | 1.6K | 5.3K | 5.9K | 7.5K | 78.2K |

Furthermore, to evaluate the scalability of our knowledge injection strategy, we randomly sample 30.1M tokens from the 76M data as a smaller training set (contributing to 0.1% of corpus ratio for knowledge injection), and also collect extra textbooks from MMedC (Qiu et al., 2024) to extend a larger training split to 132.6M tokens in total (contributing to 0.5% of corpus ratio). For the extended 132.6M corpus, the newly collected textbooks are processed as Sec. 3 states for structure-aware knowledge injection. In particular, as MMedC does not provide Japanese textbooks, we take a part of the Wikipedia data as the knowledge supplementation for the Japanese language. Here a clustering-based technique (Sarthi et al., 2024) is adopted to recursively build the knowledge structure from fragmented text segments (the others are sequential chunks from textbooks), and the processed Japanese corpus is mixed with other languages for comprehensive knowledge injection. Fig. A1 presents an example to illustrate these two kinds of knowledge structure extraction processes. Finally, we further extend the corpus ratio to 1.0%, where the newly-added data mainly comes from Japanese, French, Russian, and Spanish to balance the multi-lingual corpora.

**Data for Supervised Fine-Tuning.** As introduced in Sec. 3.3, we prompt Llama3-70B (Dubey et al., 2024) to build the structure-aware answer explanations on top of the raw SFT samples in MMedBench's training split, and generate extra QA pairs by traversing the extracted knowledge structure from textbooks. The final quantity statistics are presented in Tab. A3.

**Knowledge Structures.** We extract the domain knowledge structure for each textbook, where Fig. 3 presents an example, and combine the medical knowledge for six languages in MMedBench. As the (S)CPT corpus for Japanese is collected from Wikipedia rather than textbooks, we derive a single knowledge structure for Japanese medicine.

**Knowledge Paths and Branches**. Fig. A2 shows an example of how we define the knowledge paths and branches of the extracted knowledge structure for SSFT data synthesis.

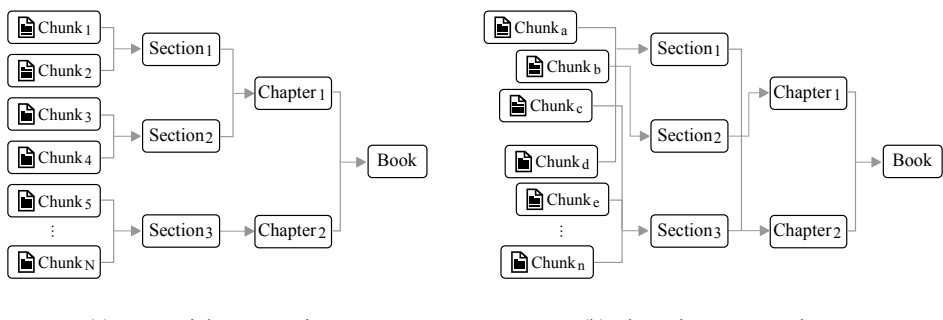

(a) Sequential Construction    (b) Clustering Construction

Figure A1: Knowledge structure extraction from (a) sequential chunks (*e.g.*, from textbooks) by our specialized 7B model and (b) separated trunks (*e.g.*, from websites) by clustering-based methods (Sarthi et al., 2024). Here the terms of "Section", "Chapter", and "book" are just examples to help illustrate the knowledge structure.

- A **path** means a knowledge path from the domain summary (*e.g.*, *Biochemistry*) to specific knowledge points (*e.g.*, *Lipid Metabolism and Cholesterol Transport*): "Biochemistry – Overview of lipoprotein metabolism, hormone synthesis – Lipoprotein Metabolism – Lipid Metabolism and Cholesterol Transport".
- A **branch** means the knowledge branch of the tree structure. If a question is related to two knowledge points (*e.g.*, *Lipid Metabolism and Cholesterol Transport* and *Pancreatic zymogen activation*) at different branches of the knowledge tree, the knowledge path contains two branches, which becomes the right-bottom part of Fig. A2.

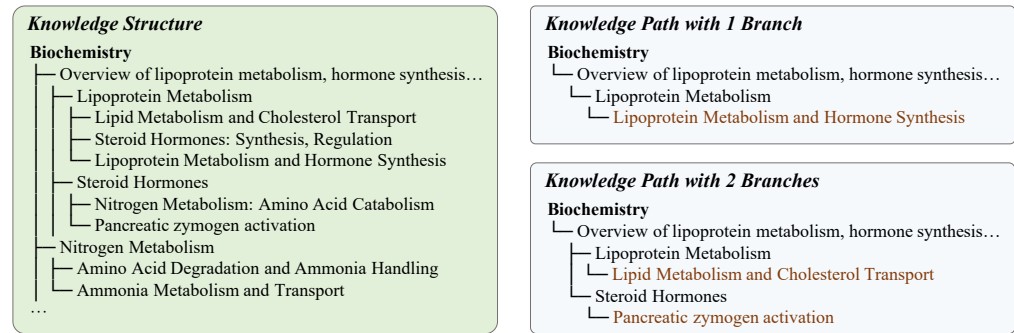

Figure A2: Definition and example of knowledge paths and branches.

### A.3    RESOURCE REQUIREMENT

We use 8 NVIDIA A100-80G GPUs to train all the models, and leverage 1-2 NVIDIA A100-80G GPUs for inference.

## B    ADDITIONAL EXPERIMENTS

### B.1    ABLATION ON KNOWLEDGE STRUCTURE EXTRACTION

Extracting domain knowledge structure is a prerequisite for subsequent knowledge injection (including both SCPT and SSFT) for language models. In Sec. 3.1, we propose a bottom-up strategy to re-chunk the texts from domain textbooks, summarize a title for each chunk, and send the title list to a specialized 7B model to derive the knowledge structure. The prompt template is displayed in Appendix C.

In fact, we argue that due to the language diversity, a perfectly recovered table of contents of textbooks is unnecessary for domain knowledge injection. A reasonable knowledge structure is sufficient enough. In Tab. A4, we individually adopt few-shot GPT-3.5-Turbo (Brown et al., 2020) and LLaMA3-70B (Dubey et al., 2024) models to extract medical knowledge structure from 18 English textbooks (Jin et al., 2020) (with 26.1M tokens) for subsequent knowledge injection (the backbone LLM is LLaMA2-7B (Touvron et al., 2023b)). Although they present 3.6%-3.7% enhancement on MMedBench (Qiu et al., 2024)'s English test set (denoted as "improvement"), leveraging GPT-3.5-Turbo and LLaMA3-70B is either expensive or time-consuming. GPT-3.5-Turbo costs around 15 dollars to process 26M tokens, while LLaMA3-70B takes around 1.5 hours on 2 A100-80G GPUs, of which both limit the scalability of data pre-processing for structure-aware knowledge injection.

Table A4: Comparison of models to extract knowledge structures on 26M English corpus.

| Model | Improvement | Time Consumption | Extra Cost |
|---|---|---|---|
| GPT-3.5-Turbo | +3.58 | - | 15$ |
| LLaMA3-70B | +3.72 | 1.5h | - |
| Ours-7B | +3.69 | **0.2h** | - |

Inspired by Liu et al. (2024a), we distilled the knowledge structure extraction capability from giant LLMs to a LLaMA2-7B model via supervised fine-tuning. In particular, we instruct LLaMA3-70B to generate 22K training examples (pairs of raw knowledge points and extracted knowledge structures) from Wikipedia, and train a LLaMA2-7B model at a batch size of 128 and a learning rate of 2e-5 for 1 epoch. After utilizing the specialized 7B model to identify the knowledge structure in medical textbooks, as shown in Tab. A4, the results translate to comparable performance on structure-aware knowledge injection. Meanwhile, the inference cost significantly decreases to 0.3 hours, which is more scalable to handle a larger amount of domain corpus.

## B.2 VERIFICATION OF THE SCALING LAW

In Sec. 4.2 in our manuscript, we preliminarily propose a scaling law for applying our structure-aware knowledge injection approach by a variety of experiments on the corpus ratio of 0.1%, 0.3%, and 0.5%. Here, we take a step to extend the scaling law at a new corpus ratio of 1.0% (the data statistics can be found in Tab. A3). According to Tab. A5, the performance of the conventional CPT+SFT paradigm and our proposed SCPT+SSFT approach shows good consistency with the scaling law's prediction. For instance, according to our fitted scaling law of $p_s \approx -1.11(\log r)2 + 7.63 \log r + 133.0$, our SCPT+SSFT strategy can bring 74.3% improvement compared to the model trained on the whole 25.5B corpus, and the empirical experiment even shows a slightly higher enhancement of 74.8%, confirming the scalability of our proposed approach.

Table A5: Relative performance enhancement on various corpus ratios.

| CorpusRatio | 0.1% | 0.3% | 0.5% | 1.0%(predict) | 1.0%(experiment) |
|---|---|---|---|---|---|
| CPT+SFT | 6.2% | 21.3% | 28.5% | 37.9% | 37.1% |
| SCPT+SSFT | 27.3% | 51.2% | 61.4% | 74.3% | 74.8% |

## B.3 COMPARISON ON TRAINING COSTS

In Tab. A6, we quantify the total training cost on 8 A100-80G GPUs. According to Qiu et al. (2024), the conventional CPT+SFT paradigm on 25.5B medicine corpus takes more than 30 days to derive the SOTA MMedLM model. In our SCPT+SSFT framework, although the pre-processing (*i.e.*, knowledge structure extraction) introduces an extra 0.6 hours to process 0.3% data (around 76M tokens), the total training process only costs 4.5 hours. As suggested in Fig. 7, when 5% training data is leveraged for knowledge injection to achieve 100% improvement, the overall cost is limited to 3 days, much less than the CPT+SFT approach with more than a month. Those analyses further demonstrate the efficacy and efficiency of our structure-aware knowledge injection framework.

Table A6: Comparison of training costs for knowledge injection.

| Paradigm | Corpus | Improvement | Pre-process | Total |
|----------|--------|-------------|-------------|-------|
| CPT+SFT | 100% | 100% | - | >30d |
| SCPT+SSFT | 0.3% | 50% | 0.6h | 4.5h |
| SCPT+SSFT | 5% | 100% | 9.7h | **3d** |

### B.4 ABLATION ON STRUCTURED KNOWLEDGE INJECTION

During the Structure-aware Continual Pre-Training (SCPT) stage, we proposed to learn specific text chunks (knowledge points) in the condition of the mindmap inputs (knowledge structures), in order to relate the knowledge points to corresponding structure nodes. In this section, we conduct a series of ablation studies to investigate the design efficacy. The vanilla CPT+SFT paradigm is adopted as the comparison baseline, where the Llama2-7B model is trained with CPT and SFT data on the English subset of MMedBench, while tested on all subsets across six languages. The hyper-parameter settings follow the main experiment in our manuscript. The empirical results are presented in Tab. A7.

Table A7: Ablation studies of SCPT on MMedBench subsets. The base model is Llama2-7B.

| Adaptation | English | Chinese | Japanese | French | Russian | Spanish | **Average** |
|------------|---------|---------|----------|--------|---------|---------|-------------|
| CPT+SFT | 46.27 | 32.57 | 26.13 | 17.36 | 50.00 | 40.63 | 35.49 |
| Ours-FixTmpl1 | 48.27 | 32.86 | 20.61 | 23.70 | 56.17 | 42.56 | 37.36 |
| Ours-FixTmpl2 | 48.10 | 32.99 | 21.23 | 23.97 | 55.97 | 43.10 | 37.56 |
| Ours-RemoveL1 | 47.90 | 32.90 | 20.11 | 24.63 | 57.10 | 43.43 | 37.68 |
| Ours-RemoveL2 | 48.05 | 33.63 | 21.62 | 24.11 | 57.49 | 43.13 | 38.00 |
| Ours-RemoveL3 | 48.47 | 33.14 | 20.15 | 23.33 | 56.86 | 43.65 | 37.60 |
| Ours-NTPLoss | 48.99 | 33.15 | 20.57 | 25.31 | 56.78 | 42.94 | 37.96 |
| **Ours-Full** | **49.10** | 33.92 | 18.33 | 27.14 | 57.42 | 43.73 | **38.27** |

First, we investigate the choice of formatting template to convert the knowledge structure to a mindmap condition. In particular, we try to fix the template to convert all knowledge mindmaps for SCPT, and randomly select two templates to repeat the experiment. According to Tab. A7, fixed SCPT templates lead to inferior performance against randomly choosing the template from the diversified 20 template pool. This is consistent with Zhu & Li (2023a)'s observation, that text rewriting can provide better knowledge augmentation for large language models.

Then, we explore the impact of the extracted knowledge structure itself. In MMedBench, a 3-layer knowledge structure (follow the *chapter-section-subsection* hierarchy) is constructed for each textbook, and we respectively remove the 1st (chapter), 2nd (section), and 3rd (subsection) layer of the hierarchy during knowledge injection. As Tab. A7 shows, removing the top layer (chapter) leads to the worst performance of 47.90%, because the remaining knowledge points cannot effectively relate to each other without the organization of the top layer. On the other hand, removing the bottom layer (subsection) performs slightly better on the English subset (because of the controlled structure-information lost), but hinders the cross-language knowledge utilization on the remaining subsets (*e.g.*, 37.60% on average across six languages).

Finally, we revisit the modeling choice of the mindmap-conditioning learning. Specifically, we try to turn the conditional modeling $p(\boldsymbol{x}^k|\boldsymbol{s}^k)$ back to complete next-token prediction $p(\boldsymbol{x}^k, \boldsymbol{s}^k)$ (the next-token prediction loss is computed on mindmap condition as well). According to Tab. A7, the performance is slightly inferior to our full version of SCPT strategy (*e.g.*, 37.96% *v.s.* 38.27% on Average). Therefore, we reserve conditional modeling for our SCPT stage.

### B.5 ABLATION ON SFT DATA SYNTHESIS

In Sec. 4.2, we compared our structure-aware knowledge injection with conventional CPT+SFT paradigm on MMedBench. On its English subset, we ablated the training components of our method, and found that the newly synthesized 8K SSFT data (by traversing the extracted knowledge structure) can inspire LLMs' cross-language capability to apply the learned structured knowledge to solve practical diagnosis problems. Here, we follow Liu et al. (2024b) to randomly generate another 8K QA pairs for SFT alignment for further comparison, denoted as "SFT*". We randomly sample medical texts and instruct Llama3-70B (Dubey et al., 2024) for data synthesis, without the knowledge structure provided. Tab. A8 indicates that "SFT*" brings slight enhancement to the English test subset, but the average accuracy drops to 34.51% instead. The results further demonstrate our method's efficacy in the application of the injected, structured domain knowledge.

Table A8: Comparison of SFT data synthesis strategies on MMedBench. The backbone LLM is the same Llama2-7B model after SCPT on English textbooks.

| SFT synthesis | English | Chinese | Japanese | French | Russian | Spanish | **Average** |
|---|---|---|---|---|---|---|---|
| - | 46.50 | 32.14 | **20.10** | 18.17 | 53.91 | 39.97 | 35.13 |
| SFT* | 47.13 | 32.49 | 16.58 | 16.72 | 51.95 | 42.16 | 34.51 |
| **SSFT*** | **49.10** | **33.92** | 18.33 | **27.14** | **57.42** | **43.73** | **38.27** |

### B.6 COMPARISON WITH OTHER SOTA KNOWLEDGE INJECTION METHODS

In this section, we compare two advanced knowledge injection methods to further demonstrate our StructTuning's efficacy: (1) AdaptLLM (Cheng et al., 2023): domain knowledge injection by appending reading comprehension QAs to each CPT chunk, and (2) RAFT (Zhang et al., 2024): improving LLM's robustness to domain-specific retrieval-augmented generation using noisy retrieval-augmented SFT samples. We take the English subset of MMedBench for evaluation, where we follow the setting in our manuscript to curate 26M tokens for CPT and use the 10K QA samples for SFT. The experimental results are displayed in Tab. A9.

Table A9: Comparison with state-of-the-art knowledge injection approaches on MMedBench.

| Approach | English | Chinese | Japanese | French | Russian | Spanish | **Average** |
|---|---|---|---|---|---|---|---|
| Vanilla | 46.27 | 32.57 | 26.13 | 17.36 | 50.00 | 40.63 | 35.49 |
| AdaptLLM | 46.19 | 33.80 | 20.60 | 14.15 | 53.12 | 42.34 | 35.03 |
| RAFT | 43.60 | 32.34 | 21.11 | 14.95 | 50.39 | 42.16 | 34.09 |
| **Ours** | **49.10** | 33.92 | 18.33 | 27.14 | 57.42 | 43.73 | **38.27** |

According to the experimental results, AdaptLLM (Cheng et al., 2023) brings negligible improvement in the final performance (*e.g.*, 46.79% *v.s.* 46.27% on the English subset), indicating such a chunk-level reading comprehension augmentation during CPT cannot help LLMs capture the entire structured domain knowledge. Concurrently, RAFT (Zhang et al., 2024) causes even worse performance, since the retrieval process introduces too many unrelated chunks and hurts LLM's QA judgments, especially when there exists a significant gap between user query and knowledge chunks in the medical diagnosis scenario.

### B.7 IN-DEPTH COMPARISON ON RETRIEVAL-AUGMENTED GENERATION

In Sec. 4.2, we briefly compare RAG adaptation and injection-based approaches in the MMedBench dataset, and this section provides more implementation details and further investigations on the popular retrieval-augmented generation approach.

**Experimental Settings.** On the implementation of the RAG baseline, we utilize the BAAI/bge-m3 (Chen et al., 2024) embedding model for dense retrieval, due to its state-of-the-art and multilingual semantic retrieval ability. For the experiments in Tab. 5, we take the same 26M English CPT

data as the knowledge base, re-chunk the data corpus for every 512 tokens, and retrieve top-3 related chunks as context inputs for LLM's generation process. The retrieval process is implemented using the LlamaIndex[2] framework.

**Additional Experiments.** We also conduct a variety of experiments to evaluate the hyper-parameters for the RAG baseline. As shown in Tab. A10, changing the chunk size and retrieved chunk number cannot bring any significant benefits. The core reason lies in the gap between user query and retrieved chunks. In particular, user queries contain many descriptive and quantitative sentences and numbers (such as the example in Fig. A3, "They enrolled 800 patients in the study, half of which have breast cancer".), and may even talk about an entirely new thing that has not been recorded in the knowledge base.

Table A10: Ablation on the hyper-parameter settings for the RAG baseline.

| ChunkSize | 256 | | | 512 | | | 1024 | | |
|---|---|---|---|---|---|---|---|---|---|
| RetrieveNum | 10 | 5 | 3 | 5 | 3 | 2 | 3 | 2 | 1 |
| Accuracy | 35.08 | 37.67 | 38.04 | 36.42 | **38.12** | 37.99 | 35.00 | 36.89 | 38.07 |

Furthermore, we also try to use the hybrid (dense+sparse) search strategy and larger rerank model (BAAI/bge-reranker-v2-m3 (Chen et al., 2024)) to enhance the retrieval quality. However according to the results in Tab. A11, the semantic gap between user queries and retrieved chunks still exists. Introducing the hybrid search and rerank model even gets worse performance (*e.g.*, the keyword *age* may be considered a key factor for hybrid search, but it cannot help to derive the answer of test sensitivity).

Table A11: Attempts to integrate hybrid-search and reranker models.

| Hybrid | Reranker | Accuracy |
|---|---|---|
| × | × | **38.12** |
| √ | × | 37.97 |
| × | √ | 37.52 |
| √ | √ | 37.75 |

**Conclusion.** RAG may assist in some knowledge-intensive tasks for information-seeking, but will encounter problems when there exists a significant semantic gap between user query and retrieved documents. MMedBench is a typical scenario, where LLMs are asked to derive medical diagnoses with proper reasoning according to the descriptions of patients or medical examinations. In this case, the retrieval process introduces too many unrelated chunks and hurts LLM's QA judgments. Fig. A3 provides an example where the retrieved chunks are actually unrelated to the complicated user query (the user asks about the analysis of a given research study, but the retrieved documents contain several keywords, *e.g.*, *age*, while having nothing to do with the *blood test study*.)

> **User Query**
>
> A pharmaceutical corporation is developing a research study to evaluate a novel blood test to screen for breast cancer. They enrolled 800 patients in the study, half of which have breast cancer. The remaining enrolled patients are age-matched controls who do not have the disease. Of those in the diseased arm, 330 are found positive for the test. Of the patients in the control arm, only 30 are found positive. What is this *test's sensitivity*?

> **Retrieved Chunks**
>
> Age Trial, the only randomized trial of breast cancer screening to specifically evaluate the impact of mammography in women age 40–49 years, found no statistically significant difference in breast cancer mortality for screened women versus controls after about 11 years of follow-up (relative risk 0.83; 95% confidence interval 0.66–1.04); however, <70% of women received screening in the intervention arm, potentially diluting the observed effect. A meta-analysis of eight large randomized trials showed a 15% relative reduction in mortality <70% of women received screening in the intervention arm, potentially diluting the observed effect. A meta-analysis of eight large randomized trials showed a 15% relative reduction in mortality (relative risk 0.85; 95% confidence interval 0.75–0.96) from mammography screening for women age 39–49 years after 11–20 years of follow-up ...

Figure A3: An example of retrieved document/chunk based on a given query.

## B.8 F1-Score Evaluation on LongBench

In Sec. 4.1, we mainly follow Zhu & Li (2023b) to investigate the memorization and understanding of injected knowledge by calculating the knowledge recall in models' responses. Here we report the F1-score measure over the Open-Book QA (OBQA) and Closed-Book QA (CBQA) settings for a

---

[2]https://www.llamaindex.ai/

thorough comparison. Note that here we use the vanilla question prompt to obtain concise answers, instead of the CoT prompt used in Sec. 4.1 to elicit models' memorized knowledge. The evaluated models are the same as Sec. 4.1.

In Tab. A12, we report the Open-Book QA (OBQA) baseline for Llama2-7B with passages as inputs, which shows the best performance on MultiFieldQA (MFQA) (Bai et al., 2023) and 2WikiMulti-hopQA (2Wiki) (Ho et al., 2020) subsets. Then, we establish the Closed-Book QA (CBQA) baseline by traditional CPT+SFT to inject passage contents into model parameters, and supplement the experiment of our SCPT+SSFT technique for comparison. According to the results shown in Tab. A12, CPT+SFT slightly improves the QA performance on several subsets (such as MultiFieldQA-zh (MFQAzh) (Bai et al., 2023)), while the overall F1-Score measure is still inferior to the OBQA performance. In contrast, our SCPT+SSFT approach successfully boosts the closed-book QA performance to 20.1% on average, even surpassing the open-book QA baseline of 18.7%.

The results are consistent with Sec. 4.1, which jointly demonstrate the effectiveness of structure-aware knowledge injection for large language models.

Table A12: F1 Score evaluation of Open-Book QA (OBQA) and Closed-Book QA (CBQA) tasks on the LongBench (Bai et al., 2023) dataset. The best results are marked in **bold**, and the secondary results are marked with underlines. The backbone model is Llama2-7B (Touvron et al., 2023b).

| Task | Adaptation | SingleDoc-QA | | | MultiDoc-QA | | | | Average |
|------|-----------|--------|------|--------|------|-------|-------|------|---------|
| | | Qasper | MFQA | MFQAzh | HpQA | 2Wiki | Musiq | Duzh | |
| OBQA | - | 19.2 | **36.8** | 11.9 | 25.4 | **32.8** | 9.4 | 5.2 | 18.7 |
| CBQA | CPT+SFT | 16.8 | 23.1 | 13.2 | 21.3 | 19.1 | 10.4 | 13.4 | 16.8 |
| | SCPT+SFT | 15.2 | 21.5 | 15.2 | 14.9 | 19.8 | 5.6 | 14.1 | 15.2 |
| | SCPT+SSFT | **19.7** | 23.5 | **19.5** | **26.4** | 24.1 | **12.2** | **15.4** | **20.1** |

## C   PROMPT TEMPLATE FOR KNOWLEDGE STRUCTURE EXTRACTION

Fig. A4 displays the prompt template to query our specialized 7B model to extract knowledge structure on given knowledge points, which introduces the task definition, detailed instruction, and output formats to illustrate the process.

You are a sophisticated AI expert in Natural Language Processing (NLP), with the specialized capability to deconstruct complex sentences and map their semantic structure. Your task is to analyze the given sentences to extract and represent the intrinsic semantic hierarchy systematically.

Follow this approach to ensure clarity and utility in your analysis:
1. **Comprehension**: Begin with a thorough reading to understand the overarching theme of the input sentences.
2. **Defining Scope**: Summarize the central theme to establish the scope of the semantic analysis.
3. **Aspect Breakdown**: Identify the core aspects of the discussion. For any aspect with additional layers, delineate "SubAspects" and repeat as necessary for complex structures. Each aspect or subaspect should be highly summarized and self-contained.
4. **Mapping**: Assign sentence numbers to their respective aspects or subaspects, indicating where in the text they are addressed.

Structure your analysis in a YAML format according to this template, and ensure the format is clean, well-organized, and devoid of extraneous commentary:
```yaml
Scope: <central theme summary>
Aspects:
 - AspectName: <main aspect>
   SentenceRange:
     start: <start sentence number>
     end: <end sentence number>
   SubAspects:
 - AspectName: <subaspect>
   SentenceRange:
     start: <start sentence number>
     end: <end sentence number>
   # Recursively repeat "SubAspects" structure as needed
 # Adjust "SubAspect" entries as needed
# Adjust "Aspect" entries as needed
```

Now, analyze the provided sentences with the structured analytical process, and output your analysis in the structured YAML format. NOTE: each aspect or subaspect should be highly summarized and self-contained, which covers at least two sentences, except for introduction or conclusion aspects.

## Content
```
{title_list}
```

## Analysis

Figure A4: Prompt template for knowledge structure identification.

```
{
"In the realm of `{field}`, a conceptual mindmap is depicted using a tree-like structure "
"to represent hierarchical relationships and thematic branches:\n\n"
"```\n{mindmap}\n```\n\n"
"Within this organized layout of `{field}`, the detailed subsection on `{section}` is described as:\n\n"
},
{
"The area of `{field}` unfolds into a rich and detailed structure, encapsulating a diverse array of topics and their interconnections. "
"These topics are organized in a manner that reflects their relationships and thematic relevance to one another, depicted through a
structured diagram:\n\n"
"```\n{mindmap}\n```\n\n"
"Within this elaborate organization, the concept of `{section}` serves as a detailed exploration into a specific element of `{field}`:\n\n"
},
{
"The `{field}` sector is structured through a complex network of concepts and categories, "
"as reflected in the following outlined representation:\n\n"
"```\n{mindmap}\n```\n\n"
"Zooming in on a discrete element of this intellectual landscape, the topic tagged as `{section}` "
"covers specific subject matter related to `{field}`:\n\n"
},
{
"Exploring the `{field}`, structured insights reveal a network of thematic areas. "
"The essence is captured in a concise diagram:\n\n"
"```\n{mindmap}\n```\n\n"
"A closer look at the portion labeled `{section}` unveils a segment rich in detail, contributing "
"to the broader understanding of `{field}`:\n\n"
},
{
"`{field}` encompasses a diverse array of themes, organized for clarity. "
"The visual schema below illustrates this organization:\n\n"
"```\n{mindmap}\n```\n\n"
"Investigating `{section}` furnishes insight into a selected theme within `{field}`, enriching the overall comprehension:\n\n"
},
{
"Contextualizing within the broader spectrum of `{field}`, the organizational structure is delineated as follows:\n\n"
"```\n{mindmap}\n```\n\n"
"Delving into `{section}`, an integral component of the `{field}` fabric, enriches the grasp of the thematic variety and depth.\n\n"
},
{
"Within the expansive knowledge area of `{field}`, an organizational schema is represented as:\n\n"
"```\n{mindmap}\n```\n\n"
"Exploring `{section}` reveals a critical facet of `{field}`, offering insights into its thematic diversity and detail.\n\n"
},
{
"The discipline of `{field}` is encapsulated by a series of interlinked concepts, mapped out as:\n\n"
"```\n{mindmap}\n```\n\n"
"The segment labeled `{section}` delves into a particular topic within `{field}`, "
"illuminating a slice of the broader intellectual landscape:\n\n"
},
{
"Navigating through `{field}`, one encounters a structured depiction of knowledge as illustrated below:\n\n"
"```\n{mindmap}\n```\n\n"
"Within this schema, `{section}` serves as a gateway to a distinct area of interest, "
"shedding light on specific sections of `{field}`:\n\n"
},
{
"Diving into the `{field}` landscape, a coherent outline presents itself, showcasing the interconnectedness of its themes:\n\n"
"```\n{mindmap}\n```\n\n"
"Focusing on the section of `{section}`, it serves as a focal point into nuanced exploration within the vast `{field}` territory:\n\n"
},
{
"The sphere of `{field}` unfolds as a network of insights and principles, outlined for comprehensive understanding:\n\n"
"```\n{mindmap}\n```\n\n"
"The exploration of `{section}` unveils a segment pivotal to the fabric of `{field}`, providing a perceiving lens:\n\n"
},
{
"As we chart the terrain of `{field}`, a constellation of concepts emerges, graphically represented as follows:\n\n"
"```\n{mindmap}\n```\n\n"
"Focusing on the component marked as `{section}`, we uncover layers within `{field}` that resonate with both breadth and depth, offering a
panoramic view into the diverse thought processes and methodologies encapsulated within.\n\n"
},
{
"`{field}` is organized into various key areas, as shown in the diagram below:\n\n"
"```\n{mindmap}\n```\n\n"
"`{section}` highlights a core area, integral for understanding the overall structure of `{field}`:\n\n"
},
{
"The structure of `{field}` is detailed below:\n\n"
"```\n{mindmap}\n```\n\n"
"A deeper understanding of `{field}` can be achieved by examining `{section}`, a vital element of its framework:\n\n"
},
{
"Overview of `{field}`'s foundational structure is as follows:\n\n"
"```\n{mindmap}\n```\n\n"
"Exploring `{section}` reveals its crucial role in comprehending the comprehensive schema of `{field}`:\n\n"
},
{
"`{field}` encompasses a range of interconnected topics, illustrated in the diagram below:\n\n"
"```\n{mindmap}\n```\n\n"
"The examination of `{section}` provides insight into how key concepts within `{field}` are interrelated:\n\n"
},
{
"Key elements within `{field}` can be organized as follows:\n\n"
"```\n{mindmap}\n```\n\n"
"Investigating the component of `{section}` is essential for grasping the complex dynamics in the `{field}` realm:\n\n"
},
{
"The `{field}` includes various components as detailed in the following structure:\n\n"
"```\n{mindmap}\n```\n\n"
"Focusing on `{section}` offers an opportunity to explore one of the numerous elements that comprise the `{field}`:\n\n"
},
{
"Within the scope of `{field}`, multiple dimensions unfold as depicted below:\n\n"
"```\n{mindmap}\n```\n\n"
"Delving into `{section}` contributes to a broader understanding of the diverse elements that construct the landscape of `{field}`:\n\n"
},
{
"Comprehensive knowledge of `{field}` can be achieved by examining its individual components, as depicted below:\n\n"
"```\n{mindmap}\n```\n\n"
"An exploration of `{section}` sheds light on its unique contribution to the `{field}`:\n\n"
}
```

Figure A5: Full template pool for mindmap conversion with 20 diversified templates.

