# OpenReview forum: "Structure-aware Domain Knowledge Injection for Large Language Models"
_ICLR.cc/2025/Conference — ICLR 2025 Conference Withdrawn Submission_

### Official Review · Reviewer_kHUD · 2024-11-02

**Soundness:** 2
**Presentation:** 2
**Contribution:** 2
**Rating:** 5
**Confidence:** 4

**Summary:**

The paper introduces “StructTuning”, a methodology designed to efficiently transform foundational Large Language Models (LLMs) into domain specialists. The authors are inspired by the human educational process and propose an interesting two-stage learning strategy: Structure-aware Continual Pre-Training (SCPT) and Structure-aware Supervised Fine-Tuning (SSFT). Evaluations on LongBench and MMedBench datasets demonstrate that StructTuning achieves comparable improvements to state-of-the-art models like MMedLM2, while significantly reducing training costs.

**Strengths:**

1. Significance of the Research Problem: This paper addresses the current challenges in enabling LLMs to learn specific domain knowledge. By designing the pre-training and fine-tuning strategy, it aims to improve the LLMs’ understanding and application of domain-specific knowledge, offering a approach to meet practical need.
2. Interesting Methodology: The proposed two-stage StructTuning approach ingeniously emulates human learning processes, gradually injecting structured domain knowledge in stages.

**Weaknesses:**

1. Unclear Experimental Setup: (1) In Open-ended Question Answering: For the LONGBENCH dataset, the division between the training and test sets is unclear. For example, as described in Section A.1, the authors generate a knowledge structure for each passage, but the sum of individual documents—reported as 200+200+150+200+200+200=1150—does not align with the 1350 entries stated on line 781 of this paper, raising concerns about the fairness and consistency of comparisons. Additionally, the reference to “10,476 passages” on line 349 is unclear in terms of how these passages differ from those in the appendix (line 781). Could the authors clarify the dataset composition, providing a detailed breakdown of how the 1350 entries and the 10,476 passages correspond to each other and to the original LongBench dataset? (2) The paper mentions “removing entries with over 0.5 F1-Score similarity to test samples to prevent data leakage.” This raises questions about whether the SSFT data generation process creates questions similar to those in the bechmark’s test set, and if so, whether filtering out sample with an F1 score over 0.5 is sufficient to prevent SSFT from merely serving as data augmentation by introducing similar samples to the test set. Could the authors provide more details on the SSFT data generation process, specifically how similarity to test samples is measured? Additionally, an analysis of the similarity distribution between generated and test samples could help justify the choice of a 0.5 F1-Score threshold. Futhermore, I am also curious about the results about CPT+SSFT, which can measure whether the performance improvement is just because SSFT introduced data augmentation. (3) In the Multiple-choice Tasks, the RAG approach consistently underperforms compared to other fine-tuning methods, which contrasts with findings in existing studies [1][2]. However, the paper does not provide specific settings for the RAG experiments, which could impact the reliability of these comparisons. Could the authors include detailed information about the RAG experimental setup, including the retrieval method, the number of retrieved documents, and how these documents were incorporated into the model’s decision-making process? This additional context would help clarify the reasons for the observed performance discrepancy.

[1] Ovadia, Oded, et al. “Fine-tuning or retrieval? comparing knowledge injection in LLMs.” arXiv preprint arXiv:2312.05934 (2023).

[2] Soudani, Heydar, Evangelos Kanoulas, and Faegheh Hasibi. “Fine Tuning vs. Retrieval Augmented Generation for Less Popular Knowledge.” arXiv preprint arXiv:2403.01432 (2024).

2. Inappropriate Dataset: (1) In the LONGBENCH Dataset: The average passage length in both single- and multi-document settings ranges from 3K to 18K, exceeding the 4K context limit of Llama-2. This makes it unsuitable for the baseline OBQA evaluation, as much of the content is truncated. While the intent might be to showcase the advantage of knowledge structuring on lengthy documents, the baseline’s context limitations lead to an unfair comparison, necessitating a more appropriate choice of long-context models for fair assessment. (2) In the Multiple-choice Task: The paper evaluates on the MMedBench dataset but uses MedTextBooks as the knowledge source rather than MMedC. The reason for not applying SCPT on MMedC is unclear. Comparing training costs across different datasets might not be valid, as smaller, high-quality datasets can yield better training outcomes, making this comparison potentially misleading.
3. Overly Conclusive Experimental Results: On line 468, the scaling law in Equation 3 is derived by fitting only four points from Figure 7. This limited dataset does not provide a sufficient basis for such a definitive conclusion, making the generalization seem premature. Could the authors consider collecting more data points to better validate the scaling law? Alternatively, if further data collection is not feasible, presenting it as a preliminary observation rather than a definitive conclusion would strengthen the reliability of this claim.

**Questions:**

1. In the open-ended question answering task on the LONGBENCH dataset, could you clarify how the training and test sets are divided? Specifically, is there a distinction between the “10,476 passages” mentioned in the paper and those referenced in the appendix?
2. In line 351, when you state “remove those with over 0.5 F1-Score similarity to test samples to prevent data leakage,” does this imply that similar questions in test set are generated? If so, does filtering out entries with an F1 score over 0.5 sufficiently prevent the performance boost from similar questions in test se?
3. In the multiple-choice task, RAG’s performance is consistently lower than fine-tuning methods. Could you provide specific experimental settings to help clarify why these results differ from those reported in existing literature?
4. The average passage length in LONGBENCH’s single- and multi-document tasks far exceeds the 4K context limit of Llama-2. How did you account for the impact of information truncation on baseline performance? Have you considered using models more suited for long-context settings as comparisons?
5. In the multiple-choice task, the MMedBench dataset uses MedTextBooks as the knowledge source instead of MMedC. Could you explain the reasoning behind this choice? Does comparing training costs across different datasets impact the fairness of the experiments?
6. On line 468, you derive the scaling law (Equation 3) based on only four data points from Figure 7. Is this approach sufficient to reach a generalizable conclusion? Have you considered introducing more data points to validate the universality of the formula?

---

### Official Review · Reviewer_kQdg · 2024-11-04

**Soundness:** 3
**Presentation:** 2
**Contribution:** 2
**Rating:** 6
**Confidence:** 3

**Summary:**

This paper introduces a pioneering methodology, termed StructTuning, to efficiently transform foundation Large Language Models (LLMs) into domain specialists. Refer to the educational processes of human students, particularly how structured domain knowledge from textbooks is assimilated and subsequently applied to tackle real-world challenges through specific exercises, the authors propose a novel two-stage strategy for knowledge injection and alignment: Structure-aware Continual Pre-Training (SCPT) and Structure-aware Supervised Fine-Tuning (SSFT). The ultimate method has undergone extensive evaluations across model architectures and scales, using closed-book question-answering tasks on LongBench and MMedBench datasets, and demonstrates the potential of comparable improvement against the state-of-the-art MMedLM2 on MMedBench, while significantly reducing the training costs to 5%.

**Strengths:**

1.Refer to the educational processes of human students, the authors propose a novel two-stage strategy for knowledge injection and alignment: Structure-aware Continual Pre-Training (SCPT) and Structure-aware Supervised Fine-Tuning (SSFT). This skillfully introduces structured data into continuous learning, and makes use of the characteristics of structured data to effectively improve the learning effect.

2.This paper first introduces the current challenges faced by LLM in learning domain-specific knowledge, and proposes an ingenious and refined method to effectively improve the understanding and application of LLM to domain-specific knowledge.

**Weaknesses:**

The analysis of experimental results is too arbitrary: for example, the author directly fits the regular curve of line 468 after obtaining the performance of the large language model under three experimental conditions: 0.1, 0.3, and 0.5. In the absence of sufficient experimental data, it seems impossible to draw such a universal conclusion. Such an arbitrary conclusion may cast doubt on the reliability of the conclusions of this paper.

**Questions:**

In the analysis of experimental results, like the one on line 468, are the conclusions drawn reliable? With such a small sample of results, is it possible to draw a regular curve? Can other arguments be added to strengthen the reliability of the conclusion?

---

### Official Review · Reviewer_iKfa · 2024-11-04

**Soundness:** 3
**Presentation:** 3
**Contribution:** 3
**Rating:** 5
**Confidence:** 4

**Summary:**

This paper introduces StructTuning, a novel two-stage framework designed to efficiently adapt large language models (LLMs) to specific domains by mimicking the structured learning process of humans.  Inspired by how students learn from structured textbook content and apply that knowledge through exercises, StructTuning comprises Structure-aware Continual Pre-Training (SCPT) and Structure-aware Supervised Fine-Tuning (SSFT).  In the SCPT stage, the method automatically extracts a domain knowledge taxonomy from the corpus (primarily textbooks) and trains the LLM to associate text chunks with their corresponding knowledge points within this taxonomy. This process effectively links fragmented information into a cohesive knowledge structure.  The subsequent SSFT stage focuses on applying the acquired knowledge.  It utilizes synthetically generated question-answer pairs, designed to prompt the LLM to explicitly leverage the learned knowledge structure in its reasoning and explanations.  The paper evaluates StructTuning on closed-book question-answering tasks using the LongBench and MMedBench datasets.  Results show substantial improvements in knowledge recall and application compared to conventional Continual Pre-Training (CPT) followed by Supervised Fine-Tuning (SFT).  Remarkably, StructTuning achieves these improvements using only a fraction (0.3%) of the training data required by standard methods, showcasing its potential for efficient and scalable domain adaptation in LLMs.

**Strengths:**

- Efficiency: The paper demonstrates impressive efficiency gains, achieving comparable performance with significantly less training data (0.3% compared to existing methods). The proposed scaling law hints at even greater potential for efficiency with larger datasets.

- Comprehensive Evaluation: The experiments cover different model architectures and scales, and utilize diverse datasets and tasks, including both recall-based and reasoning-based evaluations. The ablation study provides valuable insights into the contribution of individual components.

**Weaknesses:**

Commonality in Code Training: While the paper claims originality in its structure-aware approach, this technique is already prevalent in contexts where training data consists of code or hierarchical data structures. In code training, for instance, it’s common to replace long sequences with structured, layered representations that summarize relationships across hierarchies. This may limit the novelty of StructTuning’s proposed structure-aware methodology.

Scaling Law Verification: The proposed scaling law, while intriguing, relies heavily on extrapolation.  The lack of empirical validation for corpus ratios between 5% and 100% significantly undermines this claim.  While acknowledging resource constraints, exploring at least a few intermediate data points would considerably strengthen the paper.

Clarity on Knowledge Structure Extraction:  The paper would benefit from greater clarity on the knowledge structure extraction process using the specialized 7B model. Specifically, more information on training details and quantitative performance metrics, such as precision and recall against human-annotated structures, would strengthen the analysis. Additionally, including representative examples of extracted structures across different domains would illustrate the model’s effectiveness and adaptability. An ablation study could help evaluate the impact of the extracted structures on downstream tasks like multi-hop QA. For instance, removing different levels of the extracted structure (e.g., top-level categories versus deeper hierarchies) and analyzing the effect on QA performance would clarify whether certain structural elements are critical or if the performance degrades gracefully without them.

Lack of comparison with other knowledge injection methods: The paper would benefit from a more comprehensive comparison with a broader range of knowledge injection techniques beyond the primary comparisons to vanilla CPT+SFT and MMedLM. Including comparisons with methods like knowledge distillation tailored for domain adaptation, recent approaches integrating external knowledge graphs during fine-tuning, and hybrid methods combining retrieval-augmented generation with parameter-efficient fine-tuning would strengthen the paper’s claims of superiority. This expanded evaluation would provide a clearer picture of the proposed method’s advantages in terms of efficiency and domain adaptation capabilities.

Comparison with Retrieval-Based Methods: While RAG is briefly mentioned and experimented with (sec 4.2 & 5). The paper provides very limited details about how this RAG baseline was implemented.  It doesn't specify which retrieval method was used (e.g., dense or sparse retrieval and embedding model), the size and composition of the knowledge base, and any preprocessing or indexing steps applied. Additionally, details on retrieval hyperparameters—like the number of retrieved passages and any reranking strategies—would clarify the setup. A more rigorous comparison with state-of-the-art retrieval-augmented methods is needed to properly contextualize the contribution.  This is particularly important given the focus on knowledge-intensive tasks.

Small Typo:
Table A4, line 2 & line 3's 1st column.
P19, line 999, "self-conatined".

**Questions:**

What specific modifications were made to the loss function to accommodate the conditional modelling aspect of SCPT? How sensitive is SCPT performance to the choice of templates used to bridge mindmap structures and text chunks?  Were different template strategies compared? How do you control the length and complexity of the mindmaps to avoid overwhelming the model's context window?

The paper mentions removing synthetic QA pairs similar to the test set.  What similarity metric and threshold were used? How much data was filtered out?

How is the random walk algorithm used to generate the knowledge paths utilized in SSFT? What is the rationale for the path length parameter l? What is the impact of having too long path lengths?

Can you elaborate on the limitations of RAG observed in the MMedBench evaluation, providing concrete examples where knowledge injection proves superior?

---

### Official Review · Reviewer_Zu4M · 2024-11-06

**Soundness:** 2
**Presentation:** 3
**Contribution:** 3
**Rating:** 5
**Confidence:** 4

**Summary:**

- This paper proposes to inject knowledge from a domain-specific corpus into LLMs by StructTuning, a two-stage training strategy that comprises a Structure-aware Continual Pre-Training stage and a Structure-aware Supervised Fine-Tuning stage, to improve data efficiency (L075).
- The former stage extracts domain knowledge taxonomy from the corpus and trains LLMs to memorize the taxonomy (a tree-like mindmap of content) and predict text chunks given paths in the taxonomy.
- In the latter stage, the model is trained on question-answer-explanation triplets where the explanations are based on knowledge paths from the domain knowledge taxonomy.
- The authors examine the knowledge recall of domain-adapted language models on LongBench and their abilities to apply injected knowledge on MMedBench. On MMedBench, the proposed method can achieve 50% of the improved performance of the baseline knowledge injection method using only 0.3% data.

**Strengths:**

- This paper proposes a novel method that substantially improve the data efficiency and effectiveness of knowledge injection from domain-specific corpora into pretrained LLMs. The method is intuitive and has the nice feature of exploiting the structure in domain knowledge, organizing text chunks as a mindmap before letting the model learn the contents based on the mindmap.
- The paper is well-written, easy-to-follow, and detailed.
- The main experimental results are strong and the ablation demonstrates the helpfulness of both training stages.

**Weaknesses:**

The Questions below need to be addressed. Q4 about corpus size is important and needs clarification. Otherwise the paper is in a good shape overall.

After Author Rebuttal: While I appreciate the author responses and my clarification questions have been mostly addressed, the Q4 remains concerning. A main result in the paper is that the proposed LLM+Ours that uses 76M tokens can achieve 50% of the improvement over the baseline LLM compared to LLM+MMed that uses 25.5B tokens. However, "Ours" uses a different knowledge source than "MMed", as Reviewer kHUD also mentioned, and it is unclear whether the performance comes from the fact that the extra knowledge source is high quality. The scaling law predicted results is also not perfectly convincing, and the authors could have run the experiment with the 5% data to show the argued 100% performance. Otherwise, I believe the paper has its merits.

**Questions:**

1. Is the extracted structure always a tree? How do you guarantee this?
2. Figure 3: To confirm, you give the model the prompt on the left of Figure 3 and the model outputs the contents on the right, right?
3. L237: Can you provide more details? How come the prompt on the left of Figure 3 now allows the model to output natural language, instead of the structured output as shown in Figure 3 right?
4. L239: It appears that, during structure-aware continual pre-training, you will train the model to generate all the texts chunks from the raw corpus; just that you will condition the generation on the mindmap. Then how is it possible that in the MMedBench experiment, your corpus is much smaller than the raw corpus? L420: How do you convert a 25B-token corpus to 74M?
5. Figure 4: Does {section} means {title}?
6. L296: Could you define “path” and “branch”?
7. Table 1: Perhaps I missed this. How are SFT data generated?
8. L342: Maybe I missed this. For LongBench experiments, do you compute exact match of the answer? What if the model output has a long chain of thought? Or do you use model-based evaluation?
9. L350: Do you compute F1 on the QA, question, or answer tokens? It seems that there could still be semantically similar examples.
10. In Table 3, does +MMed correspond to CPT or CPT+SFT?

---

### Author Response · Authors · 2024-11-28
**Revision Uploaded**

Dear reviewers,

We would like to once again express our sincere gratitude for your time and effort in reviewing our manuscript. Your constructive comments have significantly enhanced the quality of our work. Based on your valuable feedback, we have made targeted revisions to the manuscript (highlighted in blue in the submitted version). These include:

- A more rigorous claim of contributions in the Abstract
- Additional descriptions of experimental setups in Appendix A
- New experiments in Appendix B (and corresponding cross-references in the main text), including:
    - Extension and verification of the scaling law analysis in Section B2
    - Ablation studies on structured knowledge injection in Section B4
    - Comparisons with state-of-the-art knowledge injection methods in Section B6
    - In-depth investigation on retrieval-augmented generation (RAG) in Section B7

As the discussion period is approaching its end, we are eager to confirm whether our responses and the revised manuscript have adequately addressed your concerns. If there are any remaining issues, we would be more than happy to provide further clarifications and explanations.

Thank you once again for your invaluable feedback and guidance.

Best regards,

Authors

---

### Note · Authors · 2024-12-16

I have read and agree with the venue's withdrawal policy on behalf of myself and my co-authors.